# Omega-3 versus Omega-6: Are We Underestimating the Ecological Significance of Arachidonic Acid in Aquatic Systems?

**DOI:** 10.3390/biom13050791

**Published:** 2023-05-03

**Authors:** Marvin Kiene, Alexander Wacker, Dominik Martin-Creuzburg

**Affiliations:** 1Department of Animal Ecology I, University of Bayreuth, 95447 Bayreuth, Germany; 2Zoological Institute and Museum, University of Greifswald, 17489 Greifswald, Germany; 3Department of Aquatic Ecology, Research Station Bad Saarow, BTU Cottbus-Senftenberg, 15526 Bad Saarow, Germany

**Keywords:** arachidonic acid, eicosapentaenoic acid, polyunsaturated fatty acids, membrane lipids, eicosanoids, co-limitation, Daphnia, freshwater food webs, limitation, ecology

## Abstract

The long-chain polyunsaturated fatty acids (PUFA) eicosapentaenoic acid (EPA, ω-3, or n-3) and arachidonic acid (ARA, ω-6 or n-6) are known to have distinct physiological functions, yet can both support growth and reproduction of consumers, raising the question of whether EPA and ARA are ecologically substitutable dietary resources. We explored the relative importance of EPA and ARA for the growth and reproduction of the freshwater keystone herbivore *Daphnia* in a life-history experiment. Both PUFA were supplemented in a concentration-dependent manner to a PUFA-free diet, separately and in combination (50% EPA: 50% ARA mixture). The growth-response curves obtained with EPA, ARA, and the mixture were virtually congruent and the thresholds for PUFA limitation did not differ, indicating that EPA (n-3) and ARA (n-6) were substitutable dietary resources under the applied experimental conditions. The actual requirements for EPA and ARA might change with growth conditions, e.g., under the influence of parasites or pathogens. The higher retention of ARA in *Daphnia* suggests that EPA and ARA are subject to different turnover rates, which also implies different physiological functions. Studies on the ARA requirements of *Daphnia* could provide valuable information on the presumably underestimated ecological importance of ARA in freshwater food webs.

## 1. Introduction

Polyunsaturated fatty acids (PUFA) are required for various physiological processes and are therefore indispensable for animal growth and reproduction. Animals are mostly unable to synthesize PUFA from low-molecular-weight precursors and therefore rely on an adequate dietary PUFA supply [1,2,3]. PUFA are fatty acids with more than one double bond in their hydrocarbon chain; they are often classified based on the location of the first double bond counted from the methyl end of the molecule. PUFA belonging to the omega-3 (n-3) and omega-6 (n-6) groups of PUFA have distinct physiological functions and are not convertible into each other. Two PUFA are classically considered to be essential, the C18 n-3 PUFA α-linolenic acid (ALA, 18:3n-3) and the C18 n-6 PUFA linoleic acid (LIN, 18:2n-6). The C20 n-3 PUFA eicosapentaenoic acid (EPA, 20:5n-3) and the C20 n-6 PUFA arachidonic acid (ARA, 20:4n-6) are also often classified as essential although the ability to synthesize these PUFA from their respective precursors ALA and LIN seems to be widespread among animals [3,4,5]. Whether or not EPA and ARA become limiting for a consumer likely depends on whether the rates of synthesis are sufficient to meet physiological demands. PUFAs are indispensable structural components of cell membranes and as such are involved in the regulation of vital membrane properties [6,7,8]. In addition, EPA and ARA are required as precursors for prostaglandins and other eicosanoids, which are important mediators in reproduction, the immune system, and ion transport physiology [9,10]. EPA also serves as a precursor for the C22 n-3 PUFA docosahexaenoic acid (DHA, 22:6n-3), which is important for the nervous system and cognitive performance [11]. Despite the indisputable physiological importance of PUFA [11,12,13], experimental studies providing unequivocal evidence for PUFA limitation are restricted to a limited number of consumers [14,15,16,17,18,19,20]. Linking dietary PUFA supply to consumer fitness is crucial for assessing the eco-evolutionary consequences of dietary PUFA deficiencies [4].

The freshwater keystone herbivore *Daphnia*, which plays a central role in freshwater food webs, has become a model to study PUFA requirements of invertebrates and for assessing potential consequences of dietary PUFA deficiencies for trophic interactions and food web processes. Supplementation experiments revealed that dietary PUFA deficiencies primarily constrain reproduction and thus population growth of *Daphnia,* e.g., [21,22]. In particular, EPA supplementation promotes the reproductive performance of *Daphnia* on diets deficient in C20-PUFA. However, similar effects have been obtained by supplementing ARA to C20-PUFA deficient diets [22,23,24,25], raising the question of whether EPA and ARA are to some extent substitutable dietary resources. This conclusion seems incompatible with the fact that EPA and ARA have distinct and partially opposing physiological functions [26,27]. However, the physiological functions of PUFA are diverse and complex, and we know little about how these multiple functions interact and how this eventually affects growth and reproduction, especially when dietary PUFA intake is low and PUFA-dependent physiological processes compete for the few PUFA available. The recognition that the growth and reproduction of *Daphnia* can be limited by more than just one nutrient simultaneously (co-limitation) [15,28] implies that also EPA and ARA may simultaneously limit the reproductive output of *Daphnia* on C20-PUFA-deficient diets, without directly interacting with or physiologically replacing each other. However, growth and reproduction depend on several basic physiological processes, and the finding that both EPA and ARA supplementation can partially or completely abolish the PUFA limitation of *Daphnia* suggests that both PUFA can maintain these basic physiological processes and are thus functionally substitutable.

To explore the relative significance of EPA and ARA for the growth and reproduction of *Daphnia*, we conducted a life-history experiment with *Daphnia magna* raised on a PUFA-free diet supplemented with increasing amounts of EPA and/or ARA. Both PUFA were either provided separately or in combination (50% EPA: 50% ARA mixture) and the obtained growth-response curves were used to assess and compare dietary thresholds for PUFA-limited population growth. We used liposomes loaded with either EPA or ARA as vehicles to provide EPA and/or ARA to *Daphnia*, following a well-established supplementation protocol. In addition, we analyzed the PUFA composition of the experimental animals to assess the fate of EPA and ARA within the body. 

## 2. Methods

### 2.1. Cultivation of Organisms

Stock cultures of a clone of *Daphnia magna*, originally isolated from ‘Großer Binnensee’, Germany [29], were maintained at 20 °C under a 16:8 (h:h) light-dark cycle in jars filled with 1.3 L filtered (0.2 µm pore-sized membrane filter) and aerated lake water. Each jar was stocked with 20 animals fed saturating concentrations (2 mg C L^−1^) of food, i.e., the green alga *Acutodesmus obliquus* (SAG 276-3a, Culture collection of algae, University of Göttingen, Germany). The green alga was grown in batch cultures at 20 °C in Cyano medium [30] and harvested in the late exponential growth phase. During the experiment, the animals were reared on the single-celled cyanobacterium *Synechococcus elongatus* (SAG 89.79). We used *S. elongatus* because it is non-toxic, phosphorus-rich, and well-ingested by *Daphnia* but lacks sterols and PUFA and has become a model food to study the effects of dietary lipids on *Daphnia* performance, e.g., [31,32,33]. Thus, the animals did not receive sterols or PUFA from the basal *S. elongatus* diet during the experiment. The cyanobacterium was cultured semi-continuously in aerated 5 L vessels containing 4 L of Cyano medium at a dilution rate of 0.2 day^−1^ and a light intensity of 200 µmol quanta m^−2^ s^−1^. Food suspensions were prepared daily by centrifugation and resuspension of the harvested *S. elongatus* cells in a fresh medium. Carbon concentrations of the food suspensions were estimated from photometric light extinctions (480 nm) using previously established carbon-extinction equations.

### 2.2. Experimental Setup

To compare the potential of EPA (20:5n-3) and ARA (20:4n-6) to support the growth and reproduction of *D. magna*, we conducted a life-history experiment in which juvenile *Daphnia* were reared on saturating concentrations (2 mg C L^−1^) of *S. elongatus* supplemented with increasing amounts of either EPA or ARA or a mixture of both (50% EPA and 50% ARA). The experiment was conducted at 20 °C under a 16:8 (h:h) light-dark cycle in experimental jars containing 200 mL of filtered (0.2 µm pore-sized membrane filter) lake water with a cohort of third-clutch neonates isolated from a stock culture (born within 12 h). PUFA supplementation was achieved by adding increasing amounts of EPA- or ARA-containing liposomes to the experimental jars. Eight dietary PUFA levels were provided: 0, 0.25, 0.75, 2, 4, 8, 16, and 24 µg PUFA mg C^−1^. The same total PUFA levels were used to generate the 50% EPA/50% ARA growth curve, i.e., the actual amount of EPA and ARA at each level was half that of the corresponding 100% treatment. *S. elongatus* is deficient not only in PUFA but also in sterols, resulting in a co-limitation by PUFA and sterols [15,28]. To release the animals from a simultaneous sterol limitation, we additionally supplemented each beaker with cholesterol-containing liposomes (26.3 µg mg C^−1^). The increasing carbon and phosphorus (constituents of phospholipids) supply via liposomes was accounted for by adding unloaded (i.e., sterol- and PUFA-free) control liposomes to the experimental beakers so that each beaker was provided in total with the same number of liposomes [32]. Each treatment initially consisted of six jars containing five individuals each. The animals were transferred daily into new jars containing freshly prepared water, food, and supplements. On day 5, three jars (replicates) were randomly selected and the contained animals were transferred into pre-weight aluminum boats and stored at −80 °C for subsequent dry mass determination and fatty acid analysis. The remaining three jars were further kept under the respective treatment conditions until the animals released their third-clutch offspring. The number of offspring produced in the first three reproduction cycles was recorded (fecundity) along with age at reproduction and age-specific survival for subsequent estimation of population growth rates. After the release of third-clutch offspring, animals were subjected to dry mass determination and fatty acid analyses as described above. Due to the very low PUFA content of animals reared at low dietary PUFA levels, we analyzed only the data of animals reared at saturating dietary PUFA levels (i.e., 16 µg mg C^−1^). At this PUFA level, the animals released their third-clutch offspring on day 20 (±1 day), i.e., they were all at approximately the same reproductive stage. We analyzed whole animals, including gut content and, if present, eggs in the brood chamber.

### 2.3. Liposome Preparation

Liposome stock suspensions were prepared by dissolving 3 mg 1-palmitoyl-2-oleoylphosphatidylglycerol (POPG) and 7 mg 1-palmitoyl-2oleoylphosphatidylcholin (POPC; Lipoid, Germany) in an aliquot of ethanol. Liposomes containing PUFA and cholesterol were prepared by adding 3.33 mg of PUFA or cholesterol from stock solutions in ethanol. Both PUFA (i.e., EPA and ARA) and cholesterol were purchased from Sigma-Aldrich. The resulting solutions were further processed as described in Martin-Creuzburg et al. 2009. Before the addition of liposomes to the experimental beakers, liposome stock suspensions were sonicated again (2 min).

### 2.4. Chemical Analyses

The fatty acid content of the experimental animals, the food, and the liposome suspensions was determined following a well-established protocol [22]. Briefly, animals or aliquots of the liposome suspensions were deposited in 7 mL of a mixture of dichloromethane and methanol (2:1, *v*:*v*) and stored at −20 °C. Total lipids were extracted three times from disrupted animal tissues (sonication) with dichloromethane:methanol (2:1, *v*:*v*). Pooled cell-free extracts were evaporated to dryness using nitrogen. The lipid extracts were esterified with 3 mol L^−1^ methanolic HCl (60 °C, 15 min). Subsequently, fatty acid methyl esters (FAMEs) were extracted three times with 2 mL of *iso*-hexane, and the lipid-containing fraction was evaporated to dryness under nitrogen and resuspended in a volume of 10 µL *iso*-hexane. Fatty acids were analyzed via gas chromatography on an HP 6890 gas chromatograph (GC) equipped with a flame ionization detector (FID) and a DB-225 (J&W Scientific, Folsom, CA, USA, 30 m × 0.25 mm ID × 0.25 µm film) capillary column. Configurations of the GC are given elsewhere [22]. The injection volume was 1.5 µL. Fatty acids were identified by comparing their retention times and mass spectra with those of reference substances. Mass spectra were recorded with a gas chromatograph-mass spectrometer (Agilent Technologies, Boblingen, Germany, 5975C inert MSD) equipped with a fused-silica capillary column (DB-225MS, Agilent; GC-MS configurations as for FID). Spectra were recorded between 50 and 600 Dalton in the electron impact (EI) ionization mode. In total, 36 different fatty acids were considered, comprising saturated (SFA), monounsaturated (MUFA), and PUFA. The absolute amount of each fatty acid was related to the dry mass of the animals or to the carbon content of the food and liposome suspensions (300 µL dried in tin capsules at 60 °C), which was determined using an elemental analyzer (EuroEA3000, HEKAtech GmbH, Wegberg, Germany).

### 2.5. Data Analysis

Population growth rates (r) were estimated based on the first three reproduction cycles using the Euler–Lotka equation:(1)$$1=\sum_{0}^{x}l_{x}m_{x}e^{-rx}$$
where $l_{x}$ is the survivorship, $m_{x}$ is the fecundity (number of viable neonates per individual), and x is the age (in days) at each reproduction cycle.

The functional relationships between dietary PUFA content and population growth rates (r) were expressed as saturation curves using a modified Monod equation:(2)$$r=r_{0}+\frac{(r_{max}-r_{0})\cdot c}{K_{s}+c}$$
where *r*_0_ is the population growth rate obtained in the absence of dietary PUFA (d^−1^), *r_max_* is the theoretical asymptotic maximum population growth rate (d^−1^), *c* is the total dietary PUFA concentration, and *K_s_* is the half-saturation constant (µg mg C^−1^). From the fitted growth curves, thresholds for PUFA-limited growth were estimated following Sperfeld and Wacker (2011) [34]. Thresholds were defined as the dietary PUFA content at which maximum population growth rates (*r_max_*) were reduced by 25%. Bootstrapping was applied to generate a parameter distribution around a median threshold, which was then used for further statistical comparison of thresholds among treatments; for details see Martin-Creuzburg et al. (2014) [32]. The body fatty acid content of the animals was analyzed using one-way ANOVA, treatment effects were analyzed using Tukey’s HSD post hoc tests with Holm’s correction method for multiple comparisons where required. Residuals were checked for normal distribution and variance homogeneity. The significance level was set to 0.05. All data analyses were performed in R version 4.1.2 [35] using the packages car, version 3.0.12 [36], and emmeans, version 1.7.2 [37].

## 3. Results

The fatty acid profile of *S. elongatus* was represented by saturated and monounsaturated fatty acids; no PUFA were detected in the *S. elongatus* food suspensions. The PUFA-limited population growth curves generated by supplementing EPA, ARA, or a mixture of both (50% each) were nearly congruent (Figure 1). Consequently, the resulting thresholds for PUFA-limited population growth (75% of *r_max_*) did not differ significantly among treatments (ANOVA, *p* = 0.956). The thresholds were 0.37, 0.36, and 0.43 µg mg C^−1^ for 100% ARA, the 50:50% mixture and 100% EPA, respectively (Figure 2).

The total SFA, MUFA, and PUFA content of the animals decreased from day 5 to day 20 of the experiment, whereas the EPA and ARA content increased (Figure 3). The decrease in total PUFA content over time was mainly caused by a decrease in C18-PUFA, i.e., ALA and LIN (Table 1). On day five of the experiment, the body fatty acid content did not differ among PUFA treatments regarding total saturated fatty acids (SFA, ANOVA, F_2,6_ = 3.16, *p* = 0.12), total monounsaturated fatty acids (MUFA, ANOVA, F_2,6_ = 1.12, *p* = 0.39) and total PUFA (ANOVA, F_2,6_ = 0.21, *p* = 0.81). On day 20 of the experiment, the total SFA, MUFA, and PUFA content of the animals still did not differ significantly among PUFA treatments (SFA: ANOVA, F_2,6_ = 1.47, *p* = 0.30; MUFA: ANOVA, F_2,6_ = 4.65, *p* = 0.06; PUFA: ANOVA, F_2,6_ = 3.62, *p* = 0.09; Figure 3).

Both EPA and ARA were detectable in animals only after supplementation, i.e., animals raised on EPA-supplemented diets contained EPA but no ARA and vice versa. On day five of the experiment, the ARA body content was significantly higher when supplied solely than when supplied in combination with EPA (Tukey’s HSD, *p* = 0.003). Similarly, the EPA body content was significantly higher when supplied solely than when supplied in combination with ARA (Tukey’s HSD, *p* = 0.004). However, although the amount of dietary EPA and ARA provided in the mixed treatment was half that provided in the respective 100% treatments, the differences in EPA and ARA body content between the mixed treatment and the respective 100% treatments did not reflect this dietary relationship, i.e., the differences were less pronounced as expected based on dietary availability. On day 20, the differences in EPA and ARA content between the mixed treatment and the respective 100% treatments even disappeared. At day 5, the total C20-PUFA content (sum of EPA and ARA) of animals in the mixed EPA/ARA treatment was higher than that of animals in the 100% EPA (Tukey’s HSD, *p* = 0.03), and the 100% ARA (not significant; Tukey’s HSD, *p* = 0.080) treatment, although the total dietary C20-PUFA content was the same in all treatments. At day 20, the total C20-PUFA content of the animals in the mixed EPA/ARA treatment was approximately twice that of animals in the 100% EPA (Tukey’s HSD, *p* = 0.07) and the 100% ARA (Tukey’s HSD, *p* = 0.09) treatment, respectively (Figure 3). In the mixed EPA/ARA treatment, the proportion of ARA within the body became higher than the proportion of EPA, statistically significant at day 20 of the experiment (Figure 3), thus not reflecting the equal dietary availability of EPA and ARA (50% each). Besides the supplemented EPA and/or ARA, no other long-chain (>C18) PUFA were detected in the animal.

## 4. Discussion

### 4.1. Substitutability of EPA and ARA

The PUFA-limited population growth responses of *Daphnia* presented here, which were obtained by supplementing a PUFA-free diet with increasing amounts of either EPA, ARA, or a mixture of both PUFA (50% each), were nearly congruent, suggesting that EPA and ARA are functionally substitutable dietary resources. Consequently, the dietary thresholds for PUFA-limited population growth did not differ statistically among treatments. The threshold for EPA-limited population growth we report here is at the lower range of previously published thresholds for somatic growth and egg [16,25,34,38,39,40]. Quantitative comparison of such thresholds across different studies is difficult because they are highly dependent on the studied life-history trait (e.g., somatic versus population growth), the supplementation method, temperature, and other nutrients potentially present in the food [15,22,41]. However, this does not hinder the direct comparison of thresholds within studies. The importance of dietary EPA for the growth and reproduction of *Daphnia* has been extensively studied in past decades, e.g., [39,42,43,44]. The importance of ARA, however, is unclear and often still questioned by scientists, despite accumulating evidence that dietary ARA can also improve the performance of *Daphnia* [22,23,24] and potentially other aquatic consumers, such as fish [13]. Our finding that thresholds for EPA- and ARA-limited population growth are statistically indistinguishable from each other complements previous work, reporting that also saturation thresholds for somatic growth and egg production of *Daphnia* were indistinguishable between EPA and ARA [22,24,25]. It is important to note here that, in our experiment, even the mixed approach with EPA and ARA (50% each) resulted in a threshold that is statistically indistinguishable from those obtained with 100% EPA or 100% ARA, strengthening the conclusion that EPA and ARA are substitutable dietary resources.

### 4.2. How Is It Possible That EPA and ARA Can Substitute for Each Other Even Though They Have Distinct Physiological Functions?

Based on our current knowledge, this question is not easy to answer. In terms of quantity, PUFA are probably most important as membrane components. The degree of unsaturation of membrane phospholipids, i.e., the number and position of double bonds within phospholipid–acyl chains, as well as the length of phospholipid–acyl chains can have major effects on vital membrane properties [6]. The incorporation of PUFAs into membrane phospholipids increases the fluidity and permeability of membranes because the kinks in the hydrocarbon chains caused by the double bonds reduce the interactions between adjacent phospholipid–acyl chains. EPA and ARA both consist of 20 carbon atoms but differ in the number (5 versus 4) and location (n-3 versus n-6) of double bonds in the carbon chain. The additional double bond in EPA adds another kink to the phospholipid–acyl chains, potentially increasing membrane fluidity and permeability more than ARA. The position of the double bonds within the molecule (i.e., n-3 vs. n-6) might be less important for maintaining membrane properties than the number of double bonds [45]. Despite these differences, both EPA and ARA incorporation may ensure the maintenance of vital membrane properties, thus allowing for growth and development. Restructuring the lipid composition of biological membranes is a major means by which ectotherms adapt to changing environmental temperatures, a mechanism known as homeoviscous adaptation [46,47]. In ectotherms, acclimatization to low temperatures typically leads to an increase in the proportion of unsaturated fatty acids within membrane phospholipids [6]. Therefore, one might consider that the presumed different effects of EPA and ARA on membrane properties only come into play at lower temperatures, when the maintenance of membrane fluidity becomes more important. However, it has been shown that both EPA and ARA can meet the increased PUFA requirements of *Daphnia* at lower temperatures [23], suggesting that the substitutability of EPA and ARA is independent of temperature. Thus, even if the presumed differential effects of EPA and ARA on membrane properties would be measurable at the cellular level, they do not appear to affect growth and reproduction.

In addition to their role as membrane components, EPA and ARA serve as precursors for eicosanoids, locally acting bioactive signaling molecules involved in regulating reproduction, immunity, and other physiological processes [9,10,48]. Research in mammalian models, including humans, emphasizes the role of EPA as a precursor for anti-inflammatory eicosanoids, and the role of ARA as a precursor for pro-inflammatory eicosanoids. This often leads to the conclusion that higher consumption of n-3 PUFA can protect against inflammatory diseases [26,27]. Although the findings obtained with mammals are not necessarily transferable to invertebrates, this supposedly opposing effect of EPA and ARA is difficult to reconcile with the substitutability of EPA and ARA postulated here. In invertebrates, the role of eicosanoids is much less clear than in vertebrates [49]. Of the three eicosanoid-forming metabolic pathways found in mammals, only one has been demonstrated unequivocally in invertebrates, the prostanoid pathway, giving rise to prostaglandins and possibly thromboxanes and prostacyclins [9,10,50,51]. Biosynthesis of EPA- and ARA-derived eicosanoids starts with the release of EPA and ARA from cellular and nuclear membrane phospholipids through the action of phospholipases. Once in their free form, EPA and ARA induce parallel and competing cascading reactions resulting in the formation of the various eicosanoids exerting control over many cellular functions, mainly regarding inflammation, immunity, reproduction, and the central nervous system [49]. Consequently, EPA and ARA compete for the same enzymes but give rise to different eicosanoids. Since the resulting eicosanoids have different physiological functions, EPA and ARA, in their function as eicosanoid precursors, cannot be substitutable. The interplay between EPA- and ARA-derived eicosanoids modulates various physiological responses [13,52]. The initial oxygenation of EPA and ARA to generate prostanoids is mediated by cyclooxygenase (COX) or, in insects and presumably other invertebrates, a COX-like chorion peroxidase (PXT) [49,53,54].

In *Daphnia*, where both *Cox* and *Pxt* gene homologs are present [10], only *Pxt* expression was shown to be up-regulated in response to both dietary EPA and ARA supplementation as well as parasite exposure (i.e., immune challenge), which is in line with the idea of prostanoids acting in *Daphnia* reproduction and immunity [55]. The same study revealed that genes coding for proteins downstream of COX/PXT respond only moderately but differently to dietary EPA or ARA supplementation in parasite-exposed *Daphnia*, suggesting differential effects of dietary EPA and ARA on *Daphnia* immunity [55]. However, the susceptibility of *Daphnia* to the highly specific bacterial parasite (*Pasteuria ramosa*) used by Schlotz et al. (2016) [55] does not appear to be affected by EPA or ARA supplementation [56]. In contrast, *Daphnia* experimentally challenged with an opportunistic gut pathogen (*Pseudomonas* sp.) were found to benefit from dietary ARA (not EPA), as demonstrated by increased survival and reproduction upon ARA supplementation [24]. Positive effects of ARA addition on the resistance to parasites and pathogens were also observed in experiments with insects [49] and fish [13]. Immune-challenged consumers may therefore have specific requirements for EPA and ARA. Further studies are needed to assess the relative importance of EPA and ARA for eicosanoid-mediated immunity.

Eicosanoids play a regulatory role also in reproductive processes [49]. The finding that EPA and ARA supplementation can similarly improve the reproductive performance of *Daphnia* suggests a non-PUFA-specific link between eicosanoid biosynthesis and reproduction. A possible mechanistic explanation may be related to the COX-like chorion peroxidase PXT, which in insects is involved in reproductive processes, such as follicle maturation [53], and which was shown to be responsive to both dietary EPA and ARA supplementation in *Daphnia* [55]. However, it remains unclear how EPA- and ARA-derived PXT products influence reproduction. Vitellogenin, a precursor of the major egg yolk protein vitellin, is indispensable for oogenesis and thus reproduction [57]. However, experiments with *Daphnia* did not reveal a clear relationship between dietary EPA and ARA supplementation and vitellogenin gene expression [48,55,58]. In fish, dietary ARA has been reported to exert significant effects on the expression of reproduction-related genes in the gonads, liver, and brain [13]. Moreover, ARA-derived eicosanoids (PGE_2_ and PGF_2α_) have been shown to act on follicle maturation in Eurasian perch [59]. The effects of EPA-derived eicosanoids on reproduction-related processes in invertebrates are mostly unclear. Eicosanoids are involved in the regulation of a variety of physiological processes, suggesting trade-offs among processes especially at low dietary EPA and/or ARA supply [56]; such trade-offs could also be influenced by the relative dietary availability of EPA and ARA.

### 4.3. Body Fatty Acid Level Hints at Substitutability, but Also Complementarity

In addition to life-history traits, we examined the fate of dietary EPA and ARA within the body of *Daphnia* at the age of five and 20 days, i.e., shortly before maturation (deposition of eggs in the brood chamber) and after the third reproduction cycle. The total SFA, MUFA, and PUFA content (mass-fraction) of the animals decreased from day 5 to day 20 of the experiment, due to the insufficient dietary lipid supply (*S. elongatus* as food), growth, and repeated maternal investment in offspring production. The decrease in total PUFA content over time was mainly driven by a decrease in C18-PUFA, i.e., ALA and LIN. ALA and LIN may as well affect the growth and reproduction of *Daphnia* [22]. However, the body ALA and LIN content decreased similarly in all PUFA supplementation treatments, suggesting that the interpretation of the growth responses shown here was not affected by a potentially emerging C18-PUFA limitation. The decrease in ALA and LIN over time also suggests that EPA and ARA were not quantitatively retro-converted to ALA and LIN. PUFA supplementation did not significantly affect the total SFA, MUFA, and PUFA content of the animals. Besides the supplemented PUFA, i.e., EPA and/or ARA, no other long-chain PUFA were detected in the animals, suggesting that EPA and ARA were not converted to PUFA with longer carbon chains, such as DHA.

The amount of dietary EPA and ARA provided in the mixed treatment (50% each) was half that provided in the respective 100% treatments. However, the differences in EPA and ARA body content between the mixed treatment and the respective 100% treatments did not reflect this dietary proportion, i.e., the differences were less pronounced as expected based on dietary availability. At day 20, the differences in EPA and ARA content between the mixed treatment and the respective 100% treatments even disappeared statistically, suggesting that EPA and ARA are more efficiently retained in the female body at low (mixed treatment) than at high (100% treatments) dietary supply. This was also reflected in the total C20-PUFA content (sum of EPA and ARA) of the animals after 20 days, which was about twice as high in the animals of the mixed EPA/ARA treatment as in the animals of the 100% EPA and 100% ARA treatment, respectively, although the total dietary PUFA content was the same in all treatments. In animals of the mixed EPA/ARA treatment, the proportion of ARA increased relative to the proportion of EPA from day 5 to day 20 of the experiment (pie charts in Figure 3), suggesting that especially ARA is retained in female *Daphnia*. It has been reported previously that ARA is more efficiently retained in *Daphnia* than EPA [60,61]. The proportion of EPA may decrease after maturity because female *Daphnia* allocate substantial amounts of EPA into their eggs during reproduction [62]. This also implies that EPA and ARA are not entirely substitutable and that both are required, possibly in different amounts, to sustain different physiological processes. In this sense, EPA and ARA would be complementary rather than substitutable nutrients. Female *Daphnia* also allocate significant amounts of ARA into their eggs, suggesting that both PUFA, EPA, and ARA, are critical for embryonic development [62,63]. It therefore seems possible that the number of viable offspring that a female *Daphnia* can produce is quantitatively linked to the availability of dietary EPA and ARA, simply because a certain amount of EPA and ARA is required per egg to sustain the various EPA- and ARA-dependent physiological processes during embryonic development.

EPA and especially ARA have been shown to accumulate in field-collected *Daphnia* [61], more than in copepods [64,65,66], and with trophic levels in the planktonic food web [67]. Accumulation of ARA in gonads and eggs of some fish species has also been linked to higher reproductive performance, although results in fish are inconsistent in this regard [13]. The n-3/n-6 PUFA ratio of food sources and that of consumers is often used to assess nutritional quality, with higher levels of n-6 PUFA, such as ARA, considered detrimental to consumers [68]. This concept is inconsistent with our finding that EPA and ARA supplementation equally improves the performance of *Daphnia* on an otherwise PUFA-free diet, suggesting that the ratio of EPA to ARA is not a good indicator of food quality.

## 5. Conclusions

We show here that both EPA and ARA supplementation, alone and in combination, can equally improve population growth rates of *Daphnia*, resulting in statistically indistinguishable dietary thresholds for EPA- and ARA-limited growth. This suggests that EPA and ARA are substitutable dietary resources, at least under the experimental conditions applied here. The actual requirements for these two C20-PUFA might change with growth conditions. Experimental evidence suggests that both EPA and ARA can meet the increased PUFA requirements of *Daphnia* at lower temperatures [23], suggesting that the substitutability of EPA and ARA is independent of temperature. The different and sometimes opposing effects of EPA- and ARA-derived eicosanoids on animal physiology are difficult to reconcile with the idea that the two PUFAs are substitutable dietary resources. Immune-challenged *Daphnia* may have different requirements for EPA and ARA. The higher retention of ARA in *Daphnia* suggests that EPA and ARA are subject to different turnover rates, which also implies different physiological functions and that EPA and ARA are complementary rather than substitutable nutrients. Further studies on PUFA-specific turnover rates under various environmental conditions, possibly using compound-specific stable isotope analyses [5,69], may help to assess the relative importance of different dietary PUFA for the performance of aquatic consumers. *Daphnia* play a key role in freshwater ecosystems and are indispensable as a model for ecological and evolutionary studies. Improving our knowledge of the ARA requirements of *Daphnia* could therefore also provide valuable information on the ecological importance of ARA in freshwater food webs.

## Figures and Tables

**Figure 1 biomolecules-13-00791-f001:**
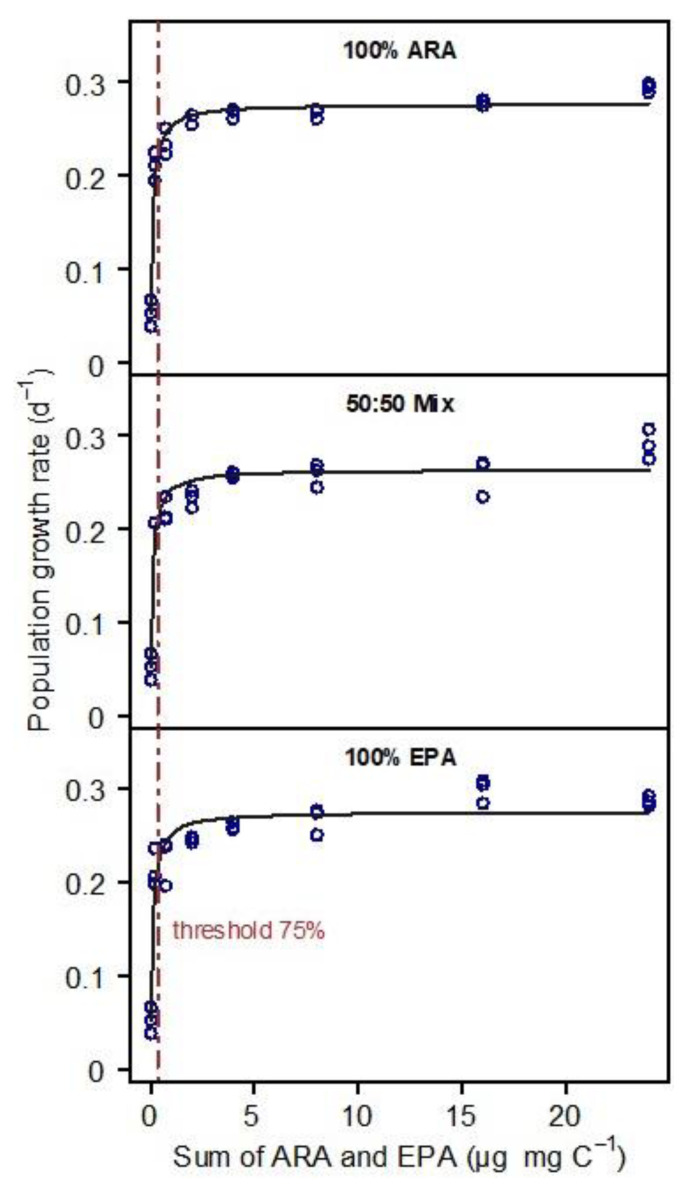
Population growth rates of *Daphnia magna* in response to increasing dietary PUFA supply (eight dietary PUFA supply levels ranging from 0 to 24 µg PUFA mg C^−1^). The growth-response curves were recorded with either ARA (100% of total dietary PUFA), EPA (100% of total dietary PUFA), or a mixture of both ARA and EPA (50% each). The *x*-axis shows the sum of both PUFA. In the 100% treatments, the sum is represented by the supplemented PUFA only, i.e., either ARA or EPA, while in the mixed treatment, the sum is represented by both PUFA (50% each). No other PUFA were detected in the food. Data points represent means of 5 animals per beaker with n = 3; lines represent fitted Monod curves. Vertical lines indicate the dietary threshold for PUFA-limited population growth (75% of *r_max_*).

**Figure 2 biomolecules-13-00791-f002:**
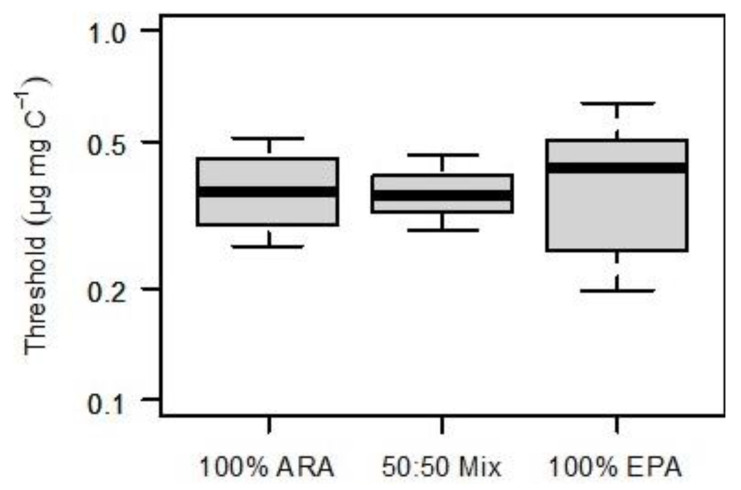
Dietary thresholds for PUFA-limited population growth (75% of *r_max_*, vertical lines in Figure 1). Either ARA (100% of total dietary PUFA), EPA (100% of total dietary PUFA), or a mixture of both, EPA and ARA (50% each) represented dietary PUFA. Bootstrapped thresholds are presented as medians (horizontal lines within each box), middle quartiles (boxes), 10%, and 90% percentiles (whiskers). Thresholds did not differ significantly among treatments (ANOVA, *p* = 0.956).

**Figure 3 biomolecules-13-00791-f003:**
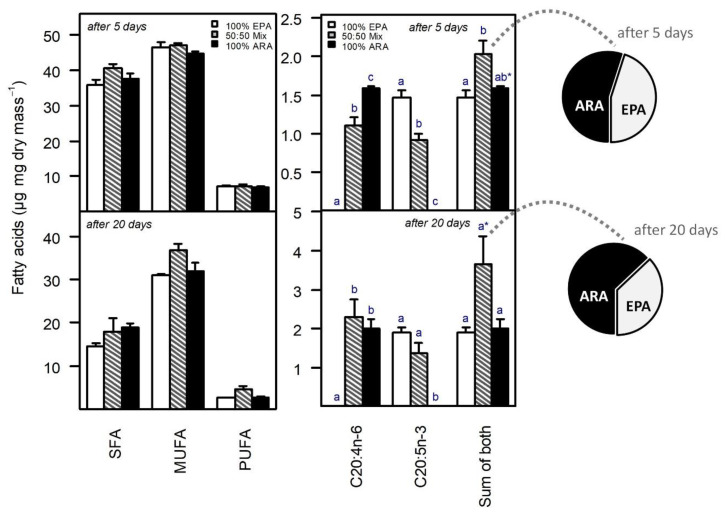
Fatty acid content of *Daphnia magna* reared on a PUFA-free diet supplemented with saturating amounts of either ARA (100% of total PUFA availability) or EPA (100% of total PUFA availability) or a mixture of ARA and EPA (50% each) at the age of five and 20 days. Total saturated (SFA), monounsaturated (MUFA), and polyunsaturated (PUFA) fatty acids on the left, ARA (20:4n-6) and EPA (20:5n-3), and the sum of both (total C20-PUFA) on the right. Pie charts on the right show the relative proportion (%) of ARA and EPA in somatic tissue of animals provided with equal amounts of dietary ARA and EPA in the mixed ARA/EPA treatment (50% each). Note the relative increase in the proportion of ARA from day 5 to day 20 from 54.9 to 62.9%. Different letters indicate significant differences among treatments for the respective group, i.e., within C20:4n-6, C20:5n-3, or the sum of both, indicated on the *x*-axis; asterisks indicate marginal significance with *p* < 0.1; data on the left do not differ significantly (ANOVA followed by Tukey’s HSD, *p* < 0.05). The total SFA, MUFA, and PUFA content decreased over time (note the different scales on the *y*-axis).

**Table 1 biomolecules-13-00791-t001:** Fatty acid content (ng mg dry mass^−1^) of *Daphnia magna* reared on a PUFA-free diet supplemented with saturating amounts of either ARA (100% of total PUFA availability) or EPA (100% of total PUFA availability) or a mixture of ARA and EPA (50% each) at the age of five and 20 days.

Day 5			
	100% EPA	50:50 Mix	100% ARA
**C13:0**	397.3 ± 48.5	528.6 ± 35.5	451.3 ± 50.8
**C14:0**	8941.2 ± 798.3	12,591.5 ± 1055.6	10,523.5 ± 1105.2
**C14:1n-7**	510.6 ± 55.3	629.2 ± 41.4	556.7 ± 52.2
**C15:0**	943.6 ± 47.0	1187.4 ± 21.1	1052.6 ± 46.9
**C16:0**	23,586.4 ± 1225.5	24,512.0 ± 458.2	23,704.7 ± 929.6
**C16:1n-7**	20,211.5 ± 1280.7	22,857.5 ± 734.5	20,961.0 ± 294.9
**C17:1n-7**	990.8 ± 84.3	990.8 ± 11.0	905.2 ± 23.8
**C18:0**	1923.0 ± 181.9	1945.4 ± 21.1	1861.6 ± 64.9
**C18:1n-9**	24,540.4 ± 1047.3	22,620.7 ± 334.2	22,494.2 ± 167.5
**C18:2n-6 (LIN)**	1104.0 ± 96.2	1030.5 ± 99.3	1012.9 ± 79.0
**C18:3n-6**	88.1 ± 124.6	218.7 ± 155.8	224.7 ± 159.4
**C18:3n-3 (ALA)**	4079.0 ± 247.4	3487.2 ± 481.0	3632.2 ± 312.6
**C18:4n-3**	457.5 ± 29.5	417.1 ± 33.1	388.3 ± 21.8
**C20:4n-6 (ARA)**	0.0 ± 0.0	1114.3 ± 137.7	1586.9 ± 35.7
**C20:5n-3 (EPA)**	1471.7 ± 125.1	915.9 ± 122.7	0.0 ± 0.0
**C22:6n-3 (DHA)**	0.0 ± 0.0	0.0 ± 0.0	0.0 ± 0.0
**Day 20**			
	100% EPA	50:50 Mix	100% ARA
**C13:0**	0.0 ± 0.0	214.4 ± 156.9	245.9 ± 26.8
**C14:0**	3180.2 ± 513.9	5670.2 ± 2167.9	5939.0 ± 358.6
**C14:1n-7**	165.6 ± 118.0	319.2 ± 236.5	372.0 ± 52.4
**C15:0**	536.0 ± 49.1	682.5 ± 143.8	658.4 ± 55.0
**C16:0**	9705.1 ± 733.0	10,205.9 ± 1877.7	10,989.7 ± 897.5
**C16:1n-7**	14,138.9 ± 842.8	17,924.4 ± 2783.1	16,809.0 ± 785.8
**C17:1n-7**	750.7 ± 15.0	310.2 ± 438.6	677.8 ± 91.4
**C18:0**	1093.6 ± 30.6	1188.3 ± 134.3	978.4 ± 103.8
**C18:1n-9**	15,736.4 ± 881.4	18,057.9 ± 2177.3	13,915.1 ± 1812.5
**C18:2n-6 (LIN)**	223.6 ± 14.4	230.6 ± 175.2	202.8 ± 38.7
**C18:3n-6**	0.0 ± 0.0	0.0 ± 0.0	168.6 ± 27.9
**C18:3n-3 (ALA)**	367.2 ± 28.5	595.8 ± 134.4	329.2 ± 62.0
**C18:4n-3**	60.7 ± 85.8	0.0 ± 0.0	0.0 ± 0.0
**C20:4n-6 (ARA)**	0.0 ± 0.0	2297.6 ± 633.8	2003.1 ± 326.7
**C20:5n-3 (EPA)**	1897.0 ± 176.7	1357.8 ± 390.5	0.0 ± 0.0
**C22:6n-3 (DHA)**	0.0 ± 0.0	0.0 ± 0.0	0.0 ± 0.0

## Data Availability

Data will be made available upon reasonable request.

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
