# Peer review of "Omega-3 versus Omega-6: Are We Underestimating the Ecological Significance of Arachidonic Acid in Aquatic Systems?"

_biomolecules, 2023, doi:10.3390/biom13050791_

Round 1

Reviewer 1 Report

This work is original and bring significant important information. I enjoyed reading the work and the novelty behind the study. I congratulate the authors for a good piece of work.

The study's main question is whether the long-chain polyunsaturated fatty acids (PUFA) eicosapentaenoic acid (EPA) and arachidonic acid (ARA) are ecologically substitutable dietary resources for the freshwater keystone herbivore Daphnia, particularly in terms of their importance for growth and reproduction.

The topic is pertinent in the area because it tackles the ecological importance and potential substitutability of various forms of PUFA in freshwater food webs, which may have ramifications for ecosystem dynamics. It also emphasizes the relevance of ARA, which has previously been eclipsed by the more commonly studied EPA.

The study contributes to the field by demonstrating that EPA and ARA are substitutable dietary resources for Daphnia in terms of growth and reproduction under the experimental settings investigated. This calls into question the widely held belief that EPA is the key limiting PUFA for aquatic herbivores and highlights the possible role of ARA in freshwater food webs.

The methods used are correct. Furthermore, the authors may consider giving more information regarding the PUFA sources used in future studies, as well as assessing the EPA and ARA turnover rates in Daphnia.

The findings are in accordance with the data and arguments offered, and they address the fundamental question raised. Under the precise experimental conditions tested, the study shows that EPA and ARA are substitutable dietary resources for Daphnia.

The references are appropriate.

Reviewer 2 Report

The comments to the MS “Omega-3 versus omega-6: Are we underestimating the ecological significance of arachidonic acid in aquatic systems?” Marvin Kiene, Alexander Wacker and Dominik Martin-Creuzburg

Introduction

Line 34-35. PUFAs include several families, but not only n-3 and n-6. There are n-9, n-7, n-4, n-1…

Line 53-54. The authors wrote that “experimental studies providing clear evidence for PUFA limitation of consumer growth or reproduction are scarce” however, they completely ignore data on experiments with Daphnia of groups of scientists led by Ann-Kristin Bergström and by Mikhail Gladyshev. This applies not only to the Introduction, but also to the Discussion.

Methods

Line 121. A more detailed description of the FA composition of the feed in general and S. elongatus in particular is required. According to Martin-Creuzburg et al., 2009 cyanobacteria did not contain PUFAs except for a small amount of linoleic acid. In this experiment, cyanobacteria were sources of linoleic or other PUFAs or not?

Line 137.  On day 5, the FA composition of daphnia of the same age was studied, and on day 20, individuals of different ages, including eggs?

Line 173. Why was concentration 16 μg mg C-1 chosen and not 24 μg mg C-1?

Figure 3. Why did the concentration of all fatty acids decrease on day 20 (almost 2 times) compared to day 5? It should be added to Results and Discussion. These two graphs with 5 and 20 days should be on the same x-axis, otherwise this great difference is not immediately visible.

In addition, the text does not explain why on day 5 the content of EPA and DHA was 1.5-2.5 mkg/mg dw, and the amount of PUFA was about 8 mkg/mg dw, while on day 20 the content of EPA and DHA was 2-4 mkg/mg dw, and the amount of PUFA was about 4 mkg/mg dw. The changes are caused by a decrease linoleic and alpha-linolenic acids? In this case, the authors do not take into account the effect on growth of these two C18PUFAs, which may be no less valuable for Daphnia and ensure high growth and reproduction. The experiment on the effect of EPA and ARA on the growth and reproduction of daphnia should have been carried out over a longer period of time in order to exclude the influence of C18PUFAs. Since the authors did not take into account the effect of C18PUFA, this possibility should be added to the Discussion, and a table with the full FA composition of Daphnia on days 5 and 20 of the experiment should be added to the Results.

What do the asterisk on the right side of the Figure 3 mean?

Discussion

Why is the term “somatic tissue” used in the Discussion? The Methods do not specify tissue separation. Moreover, according to the Methods it is obvious that the gut contents were also included in the analysis of Daphnia.

Reviewer 3 Report

The proposed manuscript “Omega-3 versus omega-6: Are we underestimating the ecological significance of arachidonic acid in aquatic systems?” greatly contributes to the study of polyunsaturated fatty acids and their functions, interactions, and ecological importance. The manuscript is technically sound, presented in an intelligible fashion, and written in good-quality English. Statistical analysis was well selected and performed. I did not find any crucial errors within the proposed manuscript.

After overall consideration of the manuscript quality I suggest minor revision. After following the below recommendations, the manuscript can meet the requirements of the Biomolecules journal.

1) Use abbreviations and the whole scientific names correctly within the whole manuscript. The first word in a sentence should not be abbreviated. On the other hand, sometimes, there are whole names which are not abbreviated.

2) Authors used liposomes in experiments which are important components of experiments. These components deserve to be introduced in the introduction.

3) In the conclusion, it is mentioned that temperature can influence substitutability of studied fatty acids. How did the authors reach this conclusion when the experiments were not based on a temperature gradient? Put citation(s) or clearly explain this conclusion.

Round 2

Reviewer 2 Report

Dear Editor and Authors

The Authors answered all questions and added important information, namely, the content of fatty acids in the studied daphnia, which is presented in Table 1. However, the Authors obviously made a mistake in the caption to the table. The caption is “Fatty acid content (µg mg dry mass-1 ) of Daphnia magna…” Obviously, the units of measurement are incorrect. It is impossible to have 90000 µg per mg dry mass, due to it equal to 90 mg per 1 mg dry mass (calculation for 5 day 100% EPA).  I assume that the correct would be µg per g dry mass-1. This needs to be checked.  

Best wishes